# Blood Based Biomarkers as Predictive Factors for Hyperprogressive Disease

**DOI:** 10.3390/jcm11175171

**Published:** 2022-09-01

**Authors:** Hasan Cagri Yildirim, Deniz Can Guven, Oktay Halit Aktepe, Hakan Taban, Feride Yilmaz, Serkan Yasar, Sercan Aksoy, Mustafa Erman, Saadettin Kilickap, Suayib Yalcin

**Affiliations:** 1Department of Medical Oncology, Hacettepe University Cancer Institute, Ankara 06230, Turkey; 2Department of Medical Oncology, Istinye University, Istanbul 34010, Turkey

**Keywords:** immunotherapy, hyperprogression, HPD, NLR, hypoalbuminemia

## Abstract

Purpose: With the widespread use of immunotherapy agents, we encounter treatment responses such as hyperprogression disease (HPD) that we have not seen with previous standard chemotherapy and targeted therapies. It is known that survival in patients with HPD is shorter than in patients without HPD. Therefore, it is important to know the factors that will predict HPD. We aimed to identify HPD-related factors in patients treated with immunotherapy. Methods: A total of 121 adult metastatic cancer patients treated with immunotherapy for any cancer were included. Baseline demographics, the ECOG performance status, type of tumors and baseline blood count parameters were recorded. Possible predisposing factors were evaluated with univariate and multivariate analyses. Results: The median age was 62.28 (interquartile range (IQR) 54.02–67.63) years, and the median follow-up was 12.26 (IQR 5.6–24.36) months. Renal cell carcinoma (33%) and melanoma (33.8%) were the most common diagnoses. Twenty patients (16.5%) had HPD. A high LDH level (*p*: 0.001), hypoalbuminemia (*p*: 0.016) and an NLR > 5 (*p*: 0.007) were found to be associated with hyperprogression. Sex (female vs. male, *p*: 0.114), age (>65 vs. <65, *p*: 0.772), ECOG (0 vs. 1–4, *p*: 0.480) and the line of treatment (1–5, *p*: 0.112) were not found to be associated with hyperprogression. Conclusions: In this study, we observed HPD in 16.5% of immunotherapy-treated patients and increased HPD risk in patients with a high LDH level (*p*: 0.001), hypoalbuminemia (*p*: 0.016) and an NLR > 5 (*p*: 0.007).

## 1. Introduction

Immunotherapy is a promising treatment option that has changed the treatment algorithms, especially in advanced cancer patients. It is widely used in melanoma [1], squamous and non-squamous non-small cell lung cancer (NSCLC) [2], renal cell carcinoma (RCC) [3], breast cancer [4], head and neck squamous cell carcinoma (HNSCC) [5], urothelial carcinoma [6] and Hodgkin lymphomas [7]. Response rates of 10%–30% are observed with ICIs (immune checkpoint inhibitors) [1,8]. However, some patients treated with these agents experience rapid treatment unresponsiveness and progression that cannot be defined by the RECIST criteria. This situation is considered as hyperprogressive disease (HPD). There is no universally accepted definition of hyperprogression at this time. For this reason, it is also difficult to determine the exact incidence of the phenomenon since this can vary with the definition used. Studies have shown that the survival time is short in patients with hyperprogression [9]. Therefore, it is important to identify factors that predict hyperprogression. In many different studies, predictive evaluations of hyperprogression have been made, but different results have been obtained. In our study, we aimed to determine the HPD-related factors.

## 2. Method

In our retrospective cohort study, 121 patients with any cancer subtype treated with ICI at Hacettepe University Cancer Institute, between September 2014 and July 2019, were retrospectively screened. All patients with baseline and at least one follow-up cross-sectional imaging with contrast after the first dose of immunotherapy were included. Baseline patient demographics, patient weight and height, ECOG (Eastern Cooperative Oncology Group) performance status, tumor histology, ICI types, comorbidities, regularly used drugs, baseline lactate dehydrogenase (LDH), neutrophil levels, thrombocyte levels and the NLR (neutrophil/lymphocyte ratio) were recorded together with survival data. We used an NLR cut-off of five, as it is the most frequently used and suggested to have a more robust relation with the prognosis in immunotherapy-treated patients [10].

Patients with HPD defined as RECIST (response evaluation criteria in solid tumors) progression who met at least three of the following criteria were included: a time-to-treatment failure < 2 months (time-to-treatment failure is defined as the time from the start of treatment with an ICI to ICI discontinuation for any reason); an increase of ≥50% in the sum of target lesions’ major diameters between the baseline and the first radiologic evaluation; the appearance of at least two new lesions in an organ already involved between the baseline and the first radiologic evaluation; spread of the disease to a new organ between the baseline and the first radiologic evaluation; and clinical deterioration with a decrease in the ECOG performance status ≥2 during the first 2 months of treatment [11].

The baseline characteristics were expressed with percentages, medians and interquartile ranges (IQR), wherever appropriate. The baseline characteristics of the patients were compared with chi-square and Mann–Whitney U tests. The association with hyperprogression risk and possible predisposing factors were evaluated with chi-square and Fischer’s exact tests. Multivariate analyses for clinical and laboratory parameters related to hyperprogression risk were conducted with backward binary logistic regression analysis, with a model including the clinical parameters with a *p* value of 0.50 or lower in the univariate analyses. Survival analysis, according to the presence or absence of hyperprogression and other clinical parameters, was performed via the Kaplan–Meier method and Cox regression analyses. The multivariate analyses results were expressed with hazard ratios (HR) and 95% confidence intervals (CI). The Statistical Package for Social Sciences version 20 program was used in the analyses. *p* values below 0.05 were considered statistically significant.

All procedures performed in studies involving human participants were in accordance with the ethical standards of the institutional and/or national research committee, and with the 1964 Helsinki declaration and its later amendments, or comparable ethical standards. The study was approved by the ethics committee of Hacettepe University.

## 3. Results

A total of 121 patients were included in the study. The median age was 62.28 (interquartile range (IQR) 54.02–67.63) years, and the median follow-up was 12.26 (IQR 5.6–24.36) months. The baseline characteristics of the patients are shown in Table 1. A total of 70% of the patients were male and 83% were over 65 years of age. Before treatment, 63% of the patients had an ECOG score of 0. A total of 40% of the patients had high LDH levels and 61% had low albumin. Of the patients, 33.9% were diagnosed with malignant melanoma, 33% with RCC (renal cell carcinoma) and 17% with NSCLC. Nineteen patients had other cancers. A total of 13% had metastases to more than two organs. Approximately 30% of patients had NLR scores above 5. Twenty patients (16.5%) had hyperprogression. The baseline characteristics of patients with and without hyperprogression are shown in Table 2. Those with a high LDH level (*p:* 0.001), hypoalbuminemia (*p:* 0.016), an NLR > 5 (*p:* 0.007) and an NSCLC diagnosis (0.026) are at high risk for hyperprogression.

In the binary logistic regression analyses including LDH levels, hypoalbuminemia, tumor type (NSCLC vs. other) and NLR levels as dependent variables, the LDH level was the only factor that had a significant association with hyperprogression risk (HR: 5.491 95% CI: 1.809–916.672, *p* = 0.003) (Table 3).

We found that patients with NSCLC had higher NLRs and hypoalbuminemia compared to other patients (*p* score: 0.018, 0.014, respectively) (Table 4).

Patients with hyperprogression had a shorter median progression-free survival time (1.86 months vs. 6.80 months, *p* < 0.001) and overall survival (4.06 months vs. 16.33 months, *p* < 0.001) compared to patients without hyperprogression (Figure 1).

## 4. Discussion

In this study, 16.5% of patients treated with immunotherapy had hyperprogression. Patients with hyperprogression had a shorter median progression-free survival time (1.86 months vs. 6.80 months, *p* < 0.001) and overall survival (4.06 months vs. 16.33 months, *p* < 0.001) compared to patients without hyperprogression. Therefore, it is important to know the predictive factors of hyperprogression.

HPD has been observed between 4% and 29% in previous studies [12,13]. The reason why it is detected in such a wide range is that there is no common definition of hyperprogression. Due to the inclusion of patients participating in clinical trials in the first studies, parameters such as tumor growth rate and tumor growth kinetics that required at least two imaging checks before immunotherapy were used in the definition of hyperprogression. However, with the widespread use of immunotherapy agents in first-line treatment, it is not possible to perform two imaging controls before treatment. Therefore, in our study, where we presented real-life data, we defined hyperprogression according to the criteria defined in Russo’s study [11].

In the study of Russo et al., in which patients with a NSCLC diagnosis were included, the incidence of hyperprogression was found to be 25.7%, while in our study it was found to be 33.3% in the NSCLC group. A total of 102 patients with RCC and 101 patients with urothelial cell carcinoma (UCC) were included [14]. HPD was observed in 5.7% of patients. In this trial, we detected HPD in 7.5% of patients with RCC. Champiat et al. report an incidence of hyperprogression of 9% (4/45 patients) during the treatment of melanoma with anti-PD1 in phase 1 trials. [15] In our trial, 9/41 (21.9%) patients had HPD.

Many studies have been conducted to identify predictive factors of hyperprogression. Many have produced different results, and many do not support each other. This may be caused by the different definitions of hyperprogression and the differences in the patient groups included in the study. Being over 65 years old in Champiat’s study [15], being of female gender in Kanjanapan’s study [16], the presence of EGFR, MDM2/4 and DNMT3A alterations in Kato’s study [12], more than two metastatic sites in Ferrara’s study [17], the density of myeloperoxidase myeloid cells within the tumor and low PD-L1 expression in tumor cells in Russo’s study [11], a high LDH level, liver metastasis and the presence of more than two metastatic sites in Kim’s study [18], an ECOG status >1 and the presence of liver metastases in Sasaki’s study [19], a high NLR level in Petrova’s study [20], and hypoalbuminemia in Hwang’s study [14] were found to be associated with hyperprogression. As can be seen, the results are not consistent with each other. In our study, a high LDH level, hypoalbuminemia, a NSCLC diagnosis and an NLR > 5 were found to be associated with hyperprogression.

In our other trial, we examined the differences between hyperprogressive disease and progressive disease and found that a high LDH level predicted hyperprogressive disease [21]. In another trial, a high LDH level was found to be associated with hyperprogression [18].

Like many other studies, we did not find a relationship between hyperprogression and the presence of more than two organ metastases, liver metastases, ECOG > 1 or the line of treatment at which the patient received immunotherapy. This situation supports the proposal that there is no relationship between hyperprogression and a high disease burden. The mechanism of hyperprogression has not been fully elucidated and it is thought to be immunologically based.

The limitations of our study are that it was a retrospective cohort study, different cancer subtypes were evaluated together, the number of our patients was insufficient, and there were many different subtypes in the diagnosis of cancer categorized as other. Additionally, the sex distribution differed from similar studies in the literature, possibly due to significantly higher smoking rates in men compared to women in Turkey. Due to the insufficient number of patients, subgroup analysis could not be performed.

## 5. Conclusions

In this study, 20 (16.5%) patients treated with ICIs developed HPD. In patients with an NLR > 5, hypoalbuminemia and elevated LDH, care should be taken in terms of hyperprogression.

## Figures and Tables

**Figure 1 jcm-11-05171-f001:**
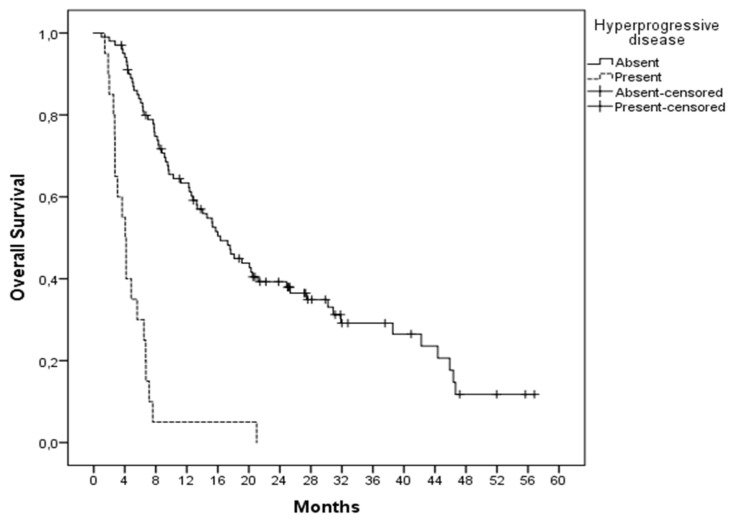
Comparison of overall survival according to the presence or absence of hyperprogression.

**Table 1 jcm-11-05171-t001:** Baseline clinical and laboratory features of patients.

		No	%
Sex	Female	36	29.7
	Male	85	70.2
Age	>65	39	32.2
	<65	82	67.7
Lactate dehydrogenase	Normal	72	59.5
	>Upper limit of normal	49	40.4
ECOG score	0	77	63.6
	1–4	36	31.8
>2 metastatic sites	Present	16	13.2
	Absent	105	86.7
Immunotherapy plus chemotherapy	Present	28	23.1
	Absent	93	76.8
Diagnosis	Melanoma	41	33.8
	RCC	40	33.0
	NSCLC	21	17.3
	Other	19	15.7
Line of treatment	1	15	12.4
	2	46	38.0
	3	27	22.3
	4	26	21.4
	5	7	5.7

**Table 2 jcm-11-05171-t002:** Baseline clinical and laboratory features of patients with or without hyperprogression.

		HPD Present	HPD Absent	*p* Score
Median age		61.6 (47.2–67.0)	62.28 (55.3–67.6)	
Sex	Female	3 (15%)	33 (32.7%)	0.091
	Male	17 (85%)	68 (67.3%)	
Age	>65	7 (35%)	32 (31.7%)	0.772
	<65	13 (65%)	69 (68.3%)	
Lactate dehydrogenase	Normal	5 (25%)	67 (66.3%)	0.001
	>ULN	15 (75%)	34 (33.7%)	
Albumin (g/dL)	>4	3 (15%)	43 (42.6%)	0.016
	<4	17 (85%)	58 (57.4%)	
ECOG score	0	11 (61.1%)	66 (69.5%)	0.485
	1–4	7 (38.9%)	29 (30.5%)	
>2 metastatic sites	Present	3 (15%)	13 (12.9%)	0.517
	Absent	17 (85%)	88 (87.1%)	
Immunotherapy plus chemotherapy	Present	6 (30%)	22 (21.8%)	0.426
	Absent	14 (70%)	79 (78.2%)	
Neutrophil-to-lymphocyte ratio	>5	11 (55%)	25 (24.7%)	0.007
	<5	9 (45%)	76 (75.7%)	
Diagnosis	Melanoma	9 (45%)	32 (31.6%)	0.026
	RCC	3 (15%)	37 (36.6%)	
	NSCLC	7 (35%)	14 (13.8%)	
	Other	1 (5%)	18 (17.8%)	
Line of treatment	1	2 (10%)	13 (12.9%)	0.112
	2	12 (60%)	34 (33.7%)	
	3	3 (15%)	24 (23.8%)	
	4	1 (5%)	25 (24.7%)	
	5	2 (10%)	5 (4.9%)	

Sex (female vs. male, *p:* 0.114), age (>65 vs. <65, *p:* 0.772), ECOG (0 vs. 1–4, *p:* 0.480), presence of liver metastases (present vs. absent, *p:* 0.752) and the line of treatment (1–5, *p:* 0.112) were not found to be associated with hyperprogression.

**Table 3 jcm-11-05171-t003:** Multivariate analysis of factors associated with hyperprogression.

Clinical Factor	Risk of Hyperprogression	
	Hazard Ratio (95%)	*p* Score
Neutrophil-to-lymphocyte ratio (>5 vs. <5)	1.972 (0.654–5.943)	0.228
Tumor type (NSCLC vs. other)	2.514 (0.752–8.411)	0.135
Albumin (low vs. normal)	3.743 (0.992–14.118)	0.051
Lactate dehydrogenase (high vs. normal)	5.491 (1.809–16.672)	0.003

**Table 4 jcm-11-05171-t004:** Lactate dehydrogenase, neutrophil-to-lymphocyte ratio and albumin status of patients with non-small cell lung cancer or others.

		Diagnosis		*p* Score
		NSCLC	Other	
Lactate dehydrogenase	Normal > ULN	11 10	61 39	0.474
Neutrophil-to-lymphocyte	>5 <5	11 10	25 75	0.018
Albumin (g/dL)	>4 <4	3 18	43 57	0.014

## Data Availability

The data used to support the findings of this study are available from the corresponding author upon request.

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
