# Peer review of "Blood Based Biomarkers as Predictive Factors for Hyperprogressive Disease"

_jcm, 2022, doi:10.3390/jcm11175171_

Round 1
Reviewer 1 Report
This study has an interesting question to address, however there are limitations in the ability to assess for predictive factors for hyperprogressive disease. It would be useful to include PD-L1 score on the biopsy specimen to help provide additional interesting data to help with prediction. Additionally, significant clinical factors of relevance such as concurrent medications are omitted from the manuscript. iRECIST criteria could be used to help provide some imaging data that would help determine if these patients had progressive disease as a result of pseudoprogression, yet died of substantial disease burden.
Author Response
Dear reviewer;
Patients evaluated as pseudoprogression in our study were not included in the hyperprogressive disease group.
Since we did not examine the PD-L1 level before immunotherapy in malignant melanoma and renal cell carcinoma, we could not examine the relationship between PD-L1 level and hyperprogressive disease.
Reviewer 2 Report
Thank you for the opportunity to review this article.
The authors report on predictors of HPD. There are several concerns.
Comment 1
In practice, HPD is problematic to distinguish from pseudo-progression, and I think there needs to be an additional note in the method as to whether pseudo-progression is included or not.
Comment 2
Predictive factors for HPD are a point to be elucidated, just as predictive factors for immunotherapy efficacy are unknown. As the authors point out, similar studies have been conducted in the previous years. I think the novelty and clinical usefulness of the present study are not well described.
Author Response
Patients evaluated as pseudoprogression according to iRECIST were included in the study, added to the group without hyperprogressive disease.
With the widespread use of immunotherapy treatments in first line treatment, the use of definitions other than Russo's criteria used in the definition of HPD will become impossible, since it is not possible to perform 2 imaging studies before IO treatment. The feature that distinguishes our study from other studies is that Russo's criteria were used.

Round 2
Reviewer 1 Report
Please edit the text to remove errors.